

# Exploring the outcomes of non-surgical periodontal therapy in modulating periodontal parameters, renal function, and inflammatory biomarkers in chronic kidney disease patients with periodontitis

Harishini Rajaratinam[1], Nurul Aliya Abdul Rahman[2], Muhammad Hafiz Hanafi[3,4], Siti Lailatul Akmar Zainuddin[5], Hanim Afzan Ibrahim[2,3,6], Muhammad Imran Kamarudin[3,4], Wan Mohd Saifuhisam Wan Zain[3,6], Sirajudeen Kuttulebbai Nainamohamed Salam[7], Salbiah Isa[8] and Nur Karyatee Kassim[2,3,6]

[1] School of Health Sciences, Universiti Sains Malaysia, Kubang Kerian, Kelantan, Malaysia
[2] School of Dental Sciences, Universiti Sains Malaysia, Kubang Kerian, Kelantan, Malaysia
[3] Hospital Pakar Universiti Sains Malaysia, Universiti Sains Malaysia, Kubang Kerian, Kelantan, Malaysia
[4] School of Medical Sciences, Universiti Sains Malaysia, Kubang Kerian, Kelantan, Malaysia
[5] Faculty of Dentistry, Manipal University College Malaysia, Bukit Baru, Melaka, Malaysia
[6] Chemical Pathology Department, School of Medical Sciences, Universiti Sains Malaysia, Kubang Kerian, Kelantan, Malaysia
[7] Department of Basic Medical Sciences, Kulliyah of Medicine, International Islamic University Malaysia, Kuantan, Pahang, Malaysia
[8] Department of Clinical Medicine, Advanced Medical and Dental Institute (IPPT), Universiti Sains Malaysia, Bertam, Pulau Pinang, Malaysia

Corresponding author
Nur Karyatee Kassim,
karyatee@usm.my

## ABSTRACT

**Background:** This comparative prospective cohort study investigated the outcomes of non-surgical periodontal therapy (NSPT) on periodontal parameters, renal function, and serum inflammatory markers in chronic kidney disease (CKD) patients with periodontitis.

**Methods:** Participants were categorised into three groups: CKD patients with periodontitis (CKD-P, $n = 20$), patients with periodontitis only (P, $n = 20$), and healthy participants (HP, $n = 20$). Periodontal parameters were initially evaluated for all participants. Blood samples were collected to assess renal function, including serum electrolytes, urea, creatinine and estimated glomerular filtration rate (eGFR), as well as inflammatory markers such as interleukin-6 (IL-6) and transforming growth factor-beta 1 (TGF-β1). NSPT was performed on both the CKD-P and P groups. Six weeks following treatment, periodontal parameters, renal function tests and inflammatory markers were re-evaluated to determine any modulation in these outcomes.

**Results:** The CKD-P group exhibited the highest concentration of potassium, urea, and creatinine. There were no significant differences in the periodontal pocket depth (PPD) and clinical attachment loss (CAL) means between CKD-P and P groups ($P > 0.05$). Similarly, there was no significant difference in the gingival bleeding index (GBI) scores between CKD-P, P, and HP groups ($P > 0.05$). However, the CKD-P

group exhibited the highest plaque score (PS) compared to the P and HP groups ($P < 0.0001$). Post-NSPT, both the CKD-P and P groups showed significant improvement in these periodontal parameters. The median eGFR for the CKD-P group improved significantly ($P < 0.0001$) after NSPT. In terms of inflammatory markers, the IL-6 levels were significantly higher in the CKD-P group compared to the P and HP groups ($P < 0.001$). Additionally, there were significant differences in the TGF-β1 levels across all three groups ($P < 0.05$). Following post-NSPT, both CKD-P ($P < 0.001$) and P ($P < 0.0001$) groups demonstrated significant reductions in IL-6. As for the TGF-β1 level, significant reduction post-NSPT was only observed in the CKD-P group ($P < 0.001$).

**Conclusion:** NSPT is effective in enhancing periodontal health, improving renal function, and decreasing systemic inflammation in CKD patients with periodontitis.

# INTRODUCTION

Kidneys play a vital role in regulating electrolytes, body fluids, and acid-base balance. Renal dysfunction can cause various electrolyte-related disorders. The Kidney Disease Outcomes Quality Initiative (KDOQI) of the National Kidney Foundation defined chronic kidney disease (CKD) as the presence of kidney damage or an estimated glomerular filtration rate (eGFR <60 ml/min per 1.73 m$^2$) for or more than 3 months (*Levey et al., 2011*). CKD is categorised into five stages according to the GFR level. Approximately 434.3 million adults in Asia are afflicted with CKD, including over 65.6 million individuals with advanced CKD (*Liyanage et al., 2022*). Oxidative stress and the buildup of advanced glycation end-products can induce inflammation in CKD (*Zoccali et al., 2017*).

Periodontitis is a chronic, non-communicable, and multifactorial disease with various etiologic and contributing factors. Periodontitis is a bacterially induced, host-mediated condition (*Kitamura et al., 2019*) characterised by the inflammation of the supporting tissues of the teeth, progressive clinical attachment loss (CAL ≥1 mm), bone loss, and formation of a pocket around the teeth and gingival recession (>3 mm). Approximately 11% of the world population is affected with severe periodontitis, and its prevalence rises as individuals grow older, reaching a plateau around the age of 50–60 years old (*Kassebaum et al., 2014*).

A cross-sectional study by *Schütz et al. (2020)* revealed that there is an association between severe periodontitis and poor renal outcomes in pre-dialytic CKD patients, thus strengthening the clinical bidirectional relationship between periodontitis and CKD. Consistent with previous findings, (*Ibrahim et al., 2020*) reported a high prevalence and severity of periodontitis among pre-dialysis CKD patients. The study revealed that, based on the periodontal pocket depth (PPD) measurements, 74% of patients exhibited mild periodontitis, 20% had moderate periodontitis, and 6% showed no clinical signs of periodontitis. In contrast, the clinical attachment loss (CAL) measurements indicated a
different distribution, with 26% of patients presenting with mild periodontitis, 63% with moderate periodontitis, and 11% with severe periodontitis (*Ibrahim et al., 2020*).

The bidirectional relationship between periodontitis and CKD is marked by several major pathophysiologic mechanisms such as impaired immune system, systemic inflammation, and oxidative stress (*Baciu, Mesaroș & Kacso, 2023*). *Porphyromonas gingivalis* pathogen of the periodontitis-provoking subgingival microbiota impairs local immune system through its ability to evade and impair the host immune-inflammatory system. *Darveau et al. (2004)* reported that the interaction of bacterial lipopolysaccharide (LPS) of *P. gingivalis* with human embryonic kidney cells involves both toll-like receptors (TLRs) 2- and 4-mediated activation of the innate immunity. However, *Coats et al. (2009)* suggests that hemin-dependent modulation of lipid A 1-dephosphorylation may alter *P. gingivalis* lipid A activity from TLR4 evasive to TLR4 suppressive. TLR activation normally results in NF-κB-mediated transcription of proinflammatory cytokines (*El-Zayat, Sibaii & Mannaa, 2019*). Moreover, CKD-induced modifications of the microenvironment also play a permissive role where both innate and adaptive immunity are impaired (*Wu et al., 2023*). A study by *Mahendra et al. (2022)* reported that the presence of periodontogenic bacteria is linked to systemic inflammation as evidenced by elevated tumour necrosis factor alpha (TNF-α) levels, which may predict the degree of kidney damage (*Lee et al., 2015*). In CKD patients, the association of periodontitis with enhanced inflammatory markers, such as C-reactive protein (CRP), has repeatedly been addressed in a systematic review by *Tavares et al. (2022)*.

Oxidative stress is one of the key mechanisms underlying the association between renal function and periodontal inflammation (*Sharma et al., 2021*). Excessive inflammation, stimulated by the immune escape mechanisms of oral microbiota *via* biofilm formation and immune suppression. This promotes the overproduction of reactive oxygen species (ROS), leading to oxidative stress. The resulting inflammatory mediators and ROS can enter the bloodstream, contributing to systemic inflammation-related diseases such as CKD. Markers of systemic oxidative stress such as 4-hydroxy-2-nonenal are associated with the severity of periodontitis (*Önder et al., 2017*). At the renal level, oxidative stress is responsible for progressive renal damage and exacerbating the ongoing systemic inflammation (*Rapa et al., 2019*). In addition, periodontitis can lead to elevated levels of endogenous inhibitors of nitric oxide synthase (NOS) (*Rapone et al., 2024*). On the other hand, the bidirectional relationship also implies that CKD may affect the oral health status of patients by triggering gingival hyperplasia, xerostomia, calcification of root canals, and delayed tooth eruption (*Wahid et al., 2013*). Inflammation in CKD is associated with various factors, including cytokine buildup due to reduced renal clearance (*Akchurin & Kaskel, 2015*). Additionally, the uremic environment in CKD patients causes oxidative stress, which worsens intestinal dysbiosis, leading to the translocation of gut bacteria into the bloodstream and triggering systemic inflammation (*Akchurin & Kaskel, 2015*).

While the exact cause-and-effect association between oral infections and systemic diseases has not been ultimately proven, there have been notable improvements in the underlying systemic disorders after treating related oral infections (*Almeida et al., 2016*; *Falcao & Bullón, 2019*). This emphasises the importance of dental health care in the

treatment of systemic disorders. Plaque-induced periodontal disease is often managed using the non-surgical periodontal therapy (NSPT). Previous studies have largely failed to explore how NSPT can simultaneously improve both periodontal health and renal function in CKD patients. The main limitations include a narrow research focus on localised periodontal outcomes rather than systemic effects, a lack of long-term studies, and the exclusion of key inflammatory and fibrotic mediators such as interleukin-6 (IL-6) and transforming growth factor-beta 1 (TGF-β1). Many studies have treated periodontitis and CKD as separate conditions, overlooking their shared inflammatory pathways. Additionally, nephrology research has underestimated the systemic benefits of periodontal therapy, leading a disconnect between the two fields. Variability in study populations, treatment protocols, and the lack of standardised guidelines further complicate the understanding of NSPT's impact on periodontal health in CKD patients. To bridge this gap, future studies must adopt a more holistic, interdisciplinary approach, evaluate relevant biomarkers, and conduct long-term investigations to establish the systemic benefits of periodontal therapy in CKD patients.

To date, most dental intervention-related studies involving peritoneal dialysis and haemodialysis patients have focused solely on the CRP level to represent systemic inflammation (*Siribamrungwong, Yothasamutr & Puangpanngam, 2014*; *Fang et al., 2015*). Therefore, IL-6 and TGF-β1 were selected due to their direct involvement in the specific pathways such as IL6/JAK/STAT3 and TGF-β1/SMAD linking periodontitis (*Aukkarasongsup et al., 2013*; *Dou et al., 2024*) and renal pathology (*Chen et al., 2019*; *Tang et al., 2021*). Both IL6/JAK/STAT3 and TGF-β1/SMAD pathways play crucial roles in the inflammatory and fibrotic responses in periodontitis and renal pathology. IL6/JAK/STAT3 primarily mediates pro-inflammatory effects, while TGF-β1/SMAD drives fibrosis. Their dysregulation can exacerbate disease progression. Focusing on these cytokine markers allows for a more targeted investigation into the mechanisms by which NSPT may influence both periodontal and renal function, in contrast to CRP which is a more general marker of inflammation and may not capture the specific cytokine-driven pathways. Hence, our objective is to explore the outcomes of NSPT in modulating periodontal parameters, renal function parameters, and concentration of inflammatory biomarkers (IL-6 and TGF-β1) in patients diagnosed both with CKD and periodontitis. We hypothesise that NSPT improves periodontal parameters, enhances renal function, and reduces the concentrations of IL-6 and TGF-β1 in CKD patients with periodontitis.

# MATERIALS AND METHODS

## Study design and ethical approval of the study

This 1-year comparative prospective cohort study was conducted at the Hospital Pakar Universiti Sains Malaysia (HPUSM), Kubang Kerian, Kelantan. It involved three groups: CKD-P ($n = 20$): CKD patients with periodontitis; P ($n = 20$): non-CKD patients with periodontitis; and HP ($n = 20$): healthy participants. The study adhered to the guidelines of the Helsinki Declaration of 1975, as revised in 2013. Ethical approval for the present study was granted by the Human Research Ethics Committee of USM (JEPeM USM code: USM/JEPeM/19100622). Sample size calculation was conducted using the software created by

*Dupont & Plummer (1997)*. To detect the mean difference of 0.32 (in the probing depth and attachment loss between CKD and non-CKD groups as reported by *Artese et al., 2010*), with a power of 80%, α = 0.05, and standard deviation (SD) = 0.5, the required sample size was 19 (considering 20% dropout rate). Therefore, the round up of 20 participants per group was determined.

## Subject inclusion and exclusion criteria

The inclusion criteria for the CKD-P and P groups include (I) aged above 18 years old, (II) exhibit good glycaemic control (HbA1c <7.0%), (III) possessed a minimum of eight teeth during the periodontal examination, (IV) diagnosed with periodontitis based on the guidelines of the 2017 World Workshop on Classification of Periodontal and Peri-Implant Diseases and Conditions (*Papapanou et al., 2018*), and (V) did not undergo scaling or root planing within the last 3 months.

An additional inclusion criterion for the CKD-P group, include diagnosed with CKD stage 3 (<60 mL/min/1.73 m$^2$) or stage 4 (<30 mL/min/1.73m$^2$) in the last 3 months before the study commenced and did not undergo dialysis. The inclusion criteria for the HP group were similar to those stated in the CKD-P and P groups, except that the participants were not diagnosed with CKD (eGFR >90 mL/min/1.73 m$^2$) and had no prior history of periodontitis. The eGFR value was calculated based on the creatinine measurements during pre-and post-NSPT, using the formula below:

GFR (mL/min/1.73 m$^2$) = 186$^*$ x Creatinine (serum)-1.154 x

X Age-0.203, X 0.742 (female) & X 1.210 (if African-American)

Subjects who had received gingival treatment prior to 1 month, patients who received any type of antibiotic within the past 2 weeks, patients who are on steroids or anti-inflammatory drugs, patients who are on immunosuppression drugs and aspirin, pregnant or lactating women and patients with medical history of rheumatic fever, congenital or valvular heart diseases or prosthetic heart valves due to possible risk of infective endocarditis with dental treatment without antibiotic coverage were excluded from the study.

## Data and sample collections

The selection of CKD-P participants involved reviewing records from the Nephrology Clinic, Hospital Pakar USM (HPUSM), with selected patients meeting the inclusion criteria contacted for their first dental visit. P group participants were identified at the dental clinic in HPUSM, while HP participants were volunteers from among hospital staff and patients' family members. All participants provided written consent and underwent a pre-NSPT phase, which included demographic data collection, dental examination, and blood sampling. Periodontal parameters such as PPD, CAL, gingival bleeding index (GBI), and plaque score (PS) were measured. NSPT, including scaling, root debridement, and oral hygiene instructions, was administered to CKD-P and P groups. After 6 weeks, the second phase involved re-evaluation of the dental parameters, renal function, and inflammatory markers to assess any modulations. Significant improvements in periodontal parameters

(*e.g.*, CAL and gingival bleeding) can be observed after 6 weeks following NSPT (*Kim, 2022*). This period is sufficient to allow for the resolution of acute inflammation and the initiation of tissue healing (*Kim, 2022*).

## Evaluation of periodontal parameters

All the periodontal examination procedures were conducted using a dental mirror, explorer, and graded William's periodontal probe with a tip diameter of 0.5 mm. The pocket depth was measured using William's periodontal probe with 1 mm incremental grading from the gingival margin to the base of the pocket depth. The probe tip was aligned with the tooth's long axis and positioned inter-proximally as close to the contact point as possible. The total measurements of PPD from the teeth involved were then averaged to produce a single mean score for each patient.

The CAL measurement was taken from the cement-enamel junction (CEJ) to the apical base of the probable crevice at the four probing sites for each tooth of the patients. The CAL was calculated by subtracting the measured distance from the free gingival margin to the CEJ from the PPD. The total measurements of CAL from the teeth involved were then averaged to produce a single mean score for each patient.

The GBI examination was performed by gently probing the orifice of the gingival crevice on four surfaces (facial/buccal, mesial, distal, palatal/lingual) of all present teeth except for the third molars. If bleeding occurred within ten seconds, a positive finding was recorded. The total number of positive sites bleeding was divided by the total number of sites examined in all present teeth and multiplied by 100%.

As for the PS examination, four surfaces of each tooth (facial/buccal, mesial, distal, palatal/lingual), excluding all third molars, were examined for the presence of plaque, and any surface with plaque presence was marked on the chart. This was divided by the total number of surfaces checked (total number of teeth present × 4) and multiplied by 100%.

The two examiners went through the inter-examiner and intra-examiner reliability and reproducibility on the measurement of PPD and CAL before the start of data collection. Repeated measurements were done on 10 different patients using a Williams periodontal probe at six sites per tooth with an agreement ± 1 mm > 90%. All measurements were rounded to the closest 0.5 mm (up or down), and when the measurement was halfway between 2 marks on the probe, the closest millimetre immediately above the mark was recorded. Each patient was examined by two examiners and after a 15- to 30-min break, the exercise was repeated (*Araujo et al., 2003*). The intraclass correlation coefficient (ICC) test was used to analyse the measured values. *Weir (2005)* stated that an ICC of 0 indicates no reliability, while the ICC of 1.0 indicates perfect reliability. He further reported that the effect of measurement error becomes minimal as interclass correlation coefficient increases above 0.80 (*Weir, 2005*).

## Provision of NSPT and oral hygiene instructions

NSPT, including scaling and root planing (SRP), followed by oral hygiene instructions were provided for groups, CKD-P and P. Full mouth SRP using an ultrasonic (EMS Piezon Master; Electro-Medical System, Nyon, Switzerland), and curettage at PPD sites with

5 mm or greater using hand scalers (Gracey, Dentsply, UK) were performed under local anaesthesia (Mepivacaine 2.2 ml with adrenaline ratio 1:100,000). All the sites were treated and irrigated with 0.2% chlorhexidine. Patients received oral hygiene instructions to brush their teeth a minimum of twice daily using fluoridated toothpaste and a soft-bristled toothbrush, and to floss once daily.

## Collection of the blood samples

About 5 ml of blood samples were collected using a serum separator tube from each participant during different visits and sent to the Chemical Pathology Laboratory, HPUSM. The blood samples were centrifuged at 3,500 rpm for 10 min. The serum was aliquoted in 1,500 μl Eppendorf tubes and kept at −80 °C.

## Estimation of renal function parameters and serum IL-6 and TGF-β1 concentrations

The serum samples were analysed spectrophotometrically for the measurements of renal function (serum electrolytes, urea, creatinine levels) using the Architect C8000 analyzer (Abbott, KC, USA). The commercial ELISA kits from *Elabscience* were used to quantitate the concentrations of IL-6 (catalogue number: E-EL-H6156) and TGF-β1 (catalogue number: E-EL-0162) in the serum. The optical densities of each well from both kits were determined using a microplate reader (Varioskan Flash Spectral scanning multimode reader; Thermo Fisher Scientific, Waltham, MA, USA) at 450 nm.

## Statistical analysis

The parametric and non-parametric analyses were determined using the outcomes from the normality test and indication based on skewness and kurtosis of data. The significance level was fixed at $P < 0.05$. Data were displayed as either mean ± standard deviation (SD) or median (interquartile range (IQR)). The error bars in the diagrams either represent the SDs (for parametric analysis) or IQR (for non-parametric analysis), depending on the selection of analyses. GraphPad Prism version 9 software was used for statistical calculations.

# RESULTS

## Sociodemographic analysis

The age differences between CKD-P, P, and HP groups were compared and analysed using the one-way ANOVA test, whereby there was a significant difference ($P < 0.0001$). The CKD-P group displayed the eldest age group, followed by the P and HP groups (Table 1).

## The renal function profile of the participants

The serum electrolytes, urea, and creatinine measurements were carried out to highlight the deteriorating renal function in patients with CKD and periodontitis, compared to those with only periodontitis and those without both diseases (Table 2). The sodium level in the CKD-P group was comparable to that of the P group. Additionally, it was evident that the CKD-P group displayed the highest potassium, urea, and creatinine concentrations, followed by the lowest calcium concentration.

**Table 1** The sociodemographic and medical characteristics of the participants.

| Sociodemographic characteristics | | Groups | | |
|---|---|---|---|---|
| | | CKD-P | P | HP |
| *Age | | [a]54.35 ± 11.73 | [a]51.35 ± 8.06 | [b]32.35 ± 7.59 |
| Gender | | | | |
| | Male, *n* (%) | 16 (80.0) | 13 (65.0) | 14 (70.0) |
| | Female, *n* (%) | 4 (20.0) | 7 (35.0) | 6 (30.0) |
| Ethnicity | | | | |
| | Malay, *n* (%) | 19 (95.00) | 19 (95.0) | 20 (100.0) |
| | Non-Malay, *n* (%) | 1 (5.00) | 1 (5.0) | Nil |
| CKD Stages | | | | |
| | Stage 3 | 8 (40.0) | Nil | Nil |
| | Stage 4 | 12 (60.0) | Nil | Nil |
| Medical illness | | | | |
| | Hypertension | 10 | 2 | Nil |
| | Diabetes | 9 | 2 | Nil |
| | Cardiovascular | 1 | Nil | Nil |
| | Others | Nil | 2 | Nil |

**Note:**
*The age-distribution of the participants was expressed in mean ± SD and analysed using the one-way ANOVA test. There was a significant difference in the mean age of the three groups ($P < 0.0001$). Tukey's multiple comparisons test was used as a *post hoc* analysis; different superscript letters indicate significant difference ($P < 0.05$).

## Clinical periodontal parameters

The independent T-test revealed no significant difference ($P > 0.05$) in PPD between the CKD-P (mean ± SD: 5.30 ± 0.83) and P (mean ± SD: 5.01 ± 0.68) groups, as shown in Fig. 1A. Similarly, there was no significant difference ($P > 0.05$) in the CAL means of CKD-P (mean ± SD: 4.71 ± 0.65) and P (mean ± SD: 3.85 ± 2.01) groups, according to the independent T-test analysis (Fig. 1B).

The mean ± SD GBI scores for the CKD-P, P, and HP groups were 51.11 ± 22.64, 45.00 ± 21.54, and 38.80 ± 22.78, respectively. The one-way ANOVA test revealed no significant difference in the GBI scores of the three groups ($P > 0.05$), as shown in Fig. 1C.

The CKD-P group exhibited the highest PS (median (IQR): 72.10 (27.67)) followed by P (median (IQR): 55.65 (42.74)) and HP (median (IQR): 4.40 (10.73)) groups. The Kruskal-Wallis test analysis revealed a significant difference in PS between the three groups ($P < 0.0001$). As shown in Fig. 1D, Dunn's multiple comparisons test was carried out as the *post hoc* test. The *post hoc* test found significant differences between the CKD-P and HP groups ($P < 0.0001$) and the P and HP groups ($P < 0.0001$). However, there was no significant difference in PS between CKD-P and P groups ($P > 0.05$).

The improvement of periodontal parameters post-NSPT was observed among the participants who were categorised under the CKD-P group (Fig. 2 & Table S1). The (A) PPD ($P < 0.0001$), (B) CAL ($P < 0.0001$), (C) GBI ($P < 0.0001$), and (D) PS ($P < 0.0001$) recorded during the pre-NSPT appointment had decreased significantly after the introduction of NSPT.

**Table 2 The renal function profile of the participants.**

| Biochemistry analytes | Groups | | | P-value |
|---|---|---|---|---|
| | CKD-P | P | HP | |
| Sodium (mmol/L) | [a]137.80 ± 2.86 | [a]137.90 ± 4.22 | [b]134.70 ± 2.41 | <0.01 |
| Urea (mmol/L) | [a]15.11 ± 7.56 | [b]6.01 ± 2.00 | [b]4.16 ± 0.80 | <0.0001 |
| Potassium (mmol/L) | [a]4.65 ± 0.71 | [b]4.18 ± 0.50 | 4.23 ± 0.54 | <0.05 |
| Calcium (mmol/L) | [a]2.21 ± 0.15 | [b]2.34 ± 0.10 | [b]2.40 ± 0.12 | <0.0001 |
| Creatinine (μmol/L) | [a]254.90 ± 78.01 | [b]87.45 ± 11.60 | [b]89.25 ± 16.03 | <0.0001 |

Note:
The data were expressed in mean ± SD. The differences (presented as P-values) in the biochemistry analytes of the CKD-P, P, and HP groups were analysed using the one-way ANOVA test. Tukey's multiple comparisons test was used as a *post hoc* analysis; different superscript letters indicate significant difference ($P < 0.05$).

The improvement of periodontal parameters following NSPT was examined among the P group participants (Fig. 3 & Table S2). The (A) PPD ($P < 0.0001$), (B) CAL ($P < 0.001$), (C) GBI ($P < 0.0001$), and (D) PS ($P < 0.0001$) recorded during the pre-NSPT appointment had declined significantly following the application of NSPT.

## The renal function test

The eGFR value of the renal function for each CKD-P participant was assessed prior to the implementation of NSPT. The median (IQR) of eGFR values during pre-NSPT was 22.60 (11.97), whereas during post-NSPT, it had increased to 26.40 (13.15) (Fig. 4). Wilcoxon's test showed a significant difference ($P < 0.0001$) by 16.81% in the eGFR value following NSPT.

## Analysis of inflammatory markers

The serum concentrations of IL-6 in the CKD-P, P, and HP groups of the current study were compared (Fig. 5A). The one-way ANOVA analysis revealed a significant difference between the three groups ($P < 0.001$). The means ± SDs for the CKD-P, P, and HP groups were 1.228 ± 0.840, 0.657 ± 0.493 and 0.408 ± 0.254, respectively. A *post hoc* Tukey's multiple comparisons tests showed significant differences in IL-6 concentrations between CKD-P and P groups ($P < 0.01$) and CKD-P and HP groups ($P < 0.001$) (Fig. 5A). However, there was no significant difference in the IL-6 levels between the P and HP groups ($P > 0.05$).

Similarly, the serum concentrations of TGF-β1 in CKD-P, P, and HP groups of the present study were compared (Fig. 5B). Following a Kruskal-Wallis analysis, it was revealed that there were significant differences among the three groups ($P < 0.05$). The medians (IQRs) of the CKD-P, P, and HP groups were 0.556 (0.832), 0.469 (0.885), and 0.206 (0.372), respectively. The CKD-P group exhibited the highest concentration of TGF-β1, followed by the P and HP groups. Following the *post-hoc* Dunn's multiple comparisons tests, it was noted that there was a significant difference in the concentration of TGF-β1 between CKD-P and HP groups ($P < 0.05$) (Fig. 5B).

The serum concentrations of IL-6 were measured in the CKD-P group, pre- and post-NSPT. Following Wilcoxon's test, it was revealed that the concentration of IL-6 in the

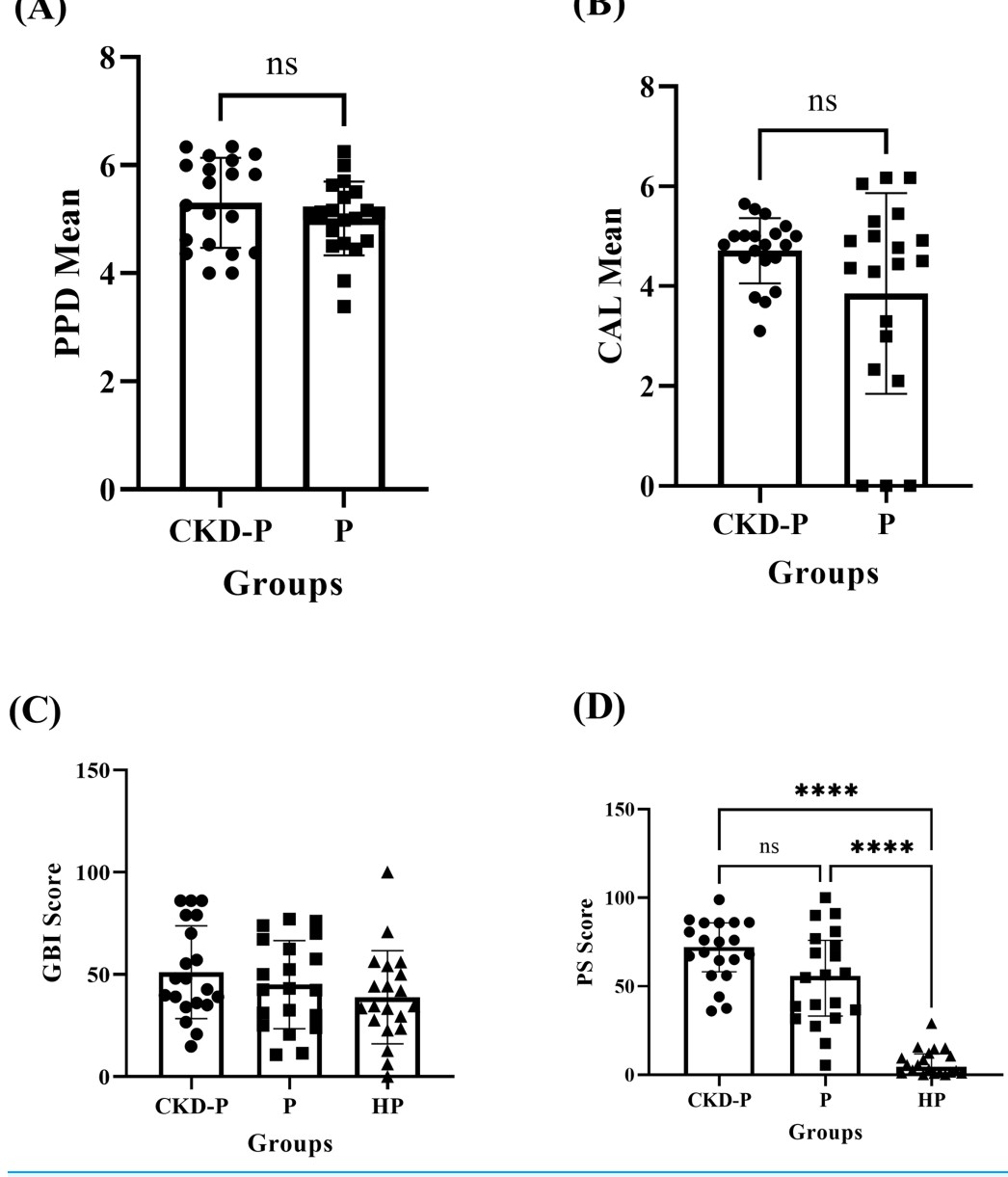

**Figure 1 The clinical periodontal parameters of CKD-P, P, and HP groups.** ns represents the non-significant difference, ****$P < 0.0001$. The (A and B) independent T-test, (C) one-way ANOVA, and (D) Kruskal-Wallis analyses were conducted accordingly. Each group consists of 20 participants and the data analysed were measured pre-NSPT.

CKD-P group was significantly reduced post-NSPT ($P < 0.001$) (Fig. 6A). The median (IQR) concentration of IL-6 levels in CKD-P patients, pre-NSPT was 1.193 (1.025) pg/ml, and it decreased by 67.14%, post-NSPT to 0.392 (0.569) pg/ml. Similarly, the concentration of IL-6 was measured in the P group, pre- and post-NSPT. Using Wilcoxon's test, it was shown that the concentration of IL-6 in the P group significantly reduced post-NSPT ($P < 0.0001$) (Fig. 6B). The median (IQR) of the concentration of IL-6
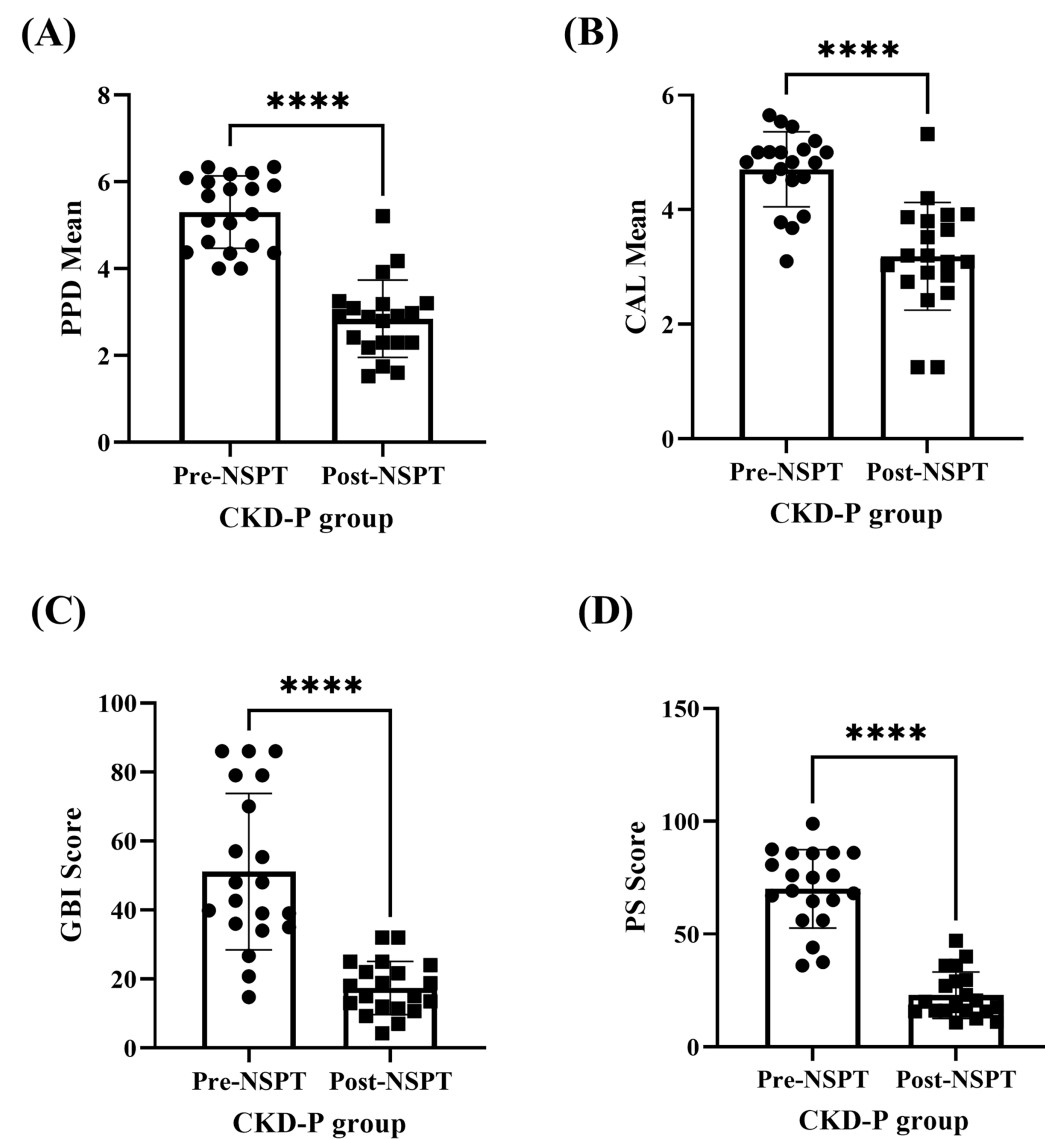

**Figure 2 The improvement in the periodontal parameters of the CKD-P group ($n = 20$) post-NSPT.** The paired T-test was conducted for analyses (A–D) and the asterisks (****) represent $P < 0.0001$.

in P patients, pre-NSPT was 0.566 (0.682) pg/ml, and it had reduced to 0.104 (0.143) pg/ml following NSPT.

The concentration of TGF-β1 was measured in the CKD-P group, pre- and post-NSPT. Using the Wilcoxon's test, it was discovered that the concentration of TGF-β1 in the CKD-P group was significantly reduced post-NSPT ($P < 0.001$) (Fig. 6C). The median (IQR) of the concentration of TGF-β1 in the serum of the CKD-P group before NSPT was 0.556 (0.832) pg/ml, and post-NSPT, it had reduced to 0.176 (0.281) pg/ml. In contrast, it was discovered that the concentration of TGF-β1 in the P group did not differ significantly following post-NSPT ($P > 0.05$). The analysis was conducted using a paired T-test

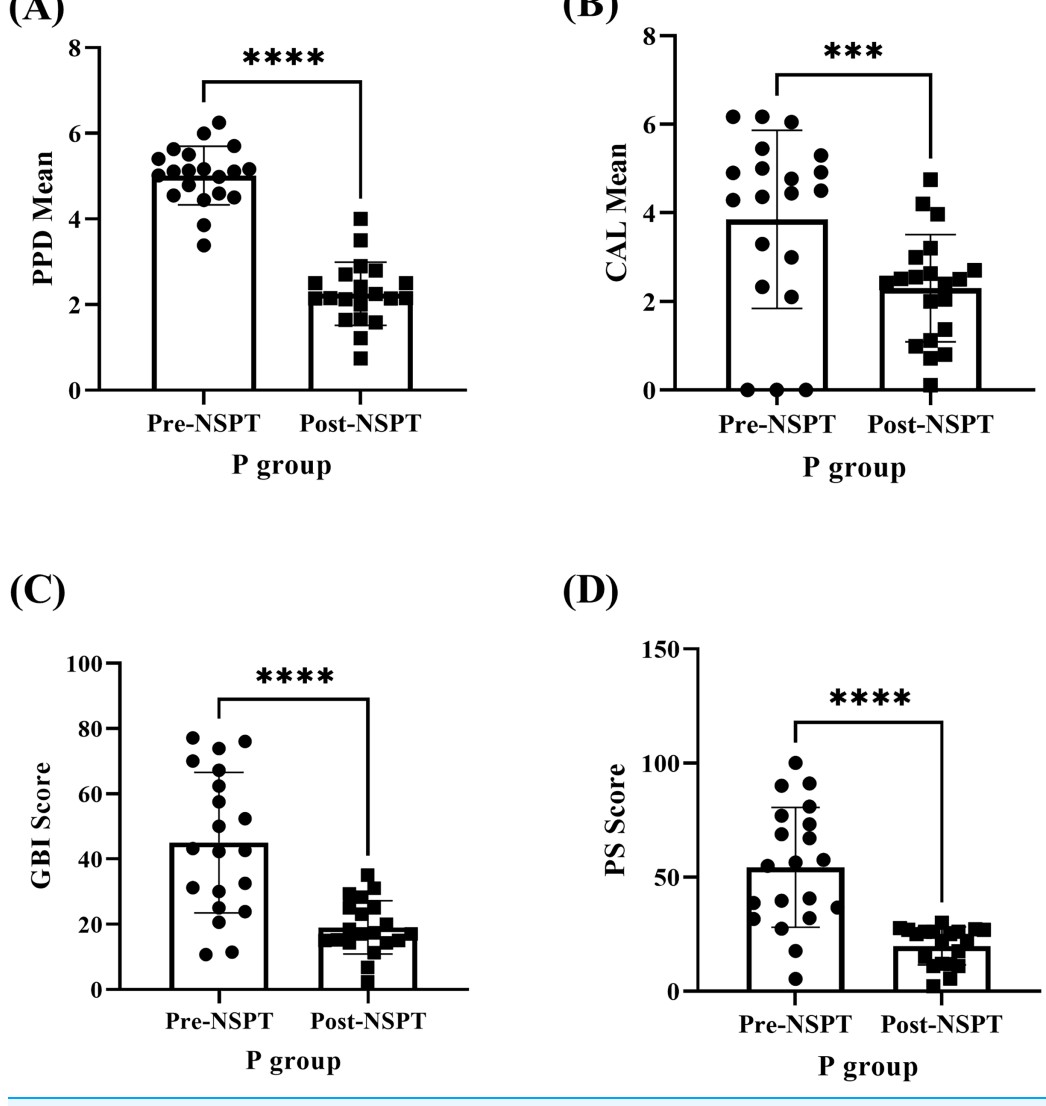

**Figure 3 The improvement in the periodontal parameters of the P group (*n* = 20) post-NSPT.** The analyses (A–D) were carried out using the paired T-test. Four asterisks (****) represent *P* < 0.0001 and three asterisks (***) represent *P* < 0.001.

(Fig. 6D). The mean ± SDs of the concentration of TGF-β1 in the serum of the P group, pre-NSPT was 0.587 ± 0.470 pg/ml, and following NSPT, it was approximately 0.562 ± 0.445 pg/ml. However, after treatment was introduced, a minute mean difference can still be observed, varying by 0.025 pg/ml.

A multiple linear regression (MLR) analysis was conducted to assess whether clinical dental parameters (PPD, CAL, GBI, and PS) and inflammatory biomarker concentrations (IL-6 and TGF-β1) could predict the rate of change in eGFR. The independent variables were represented as the differences between pre- and post-NSPT values (ΔPPD, ΔCAL, ΔGBI, ΔPS, ΔIL-6, and ΔTGF-β1). The rate of change in eGFR (dependent variable) was calculated by dividing the difference between pre- and post-NSPT eGFR by 0.115 year

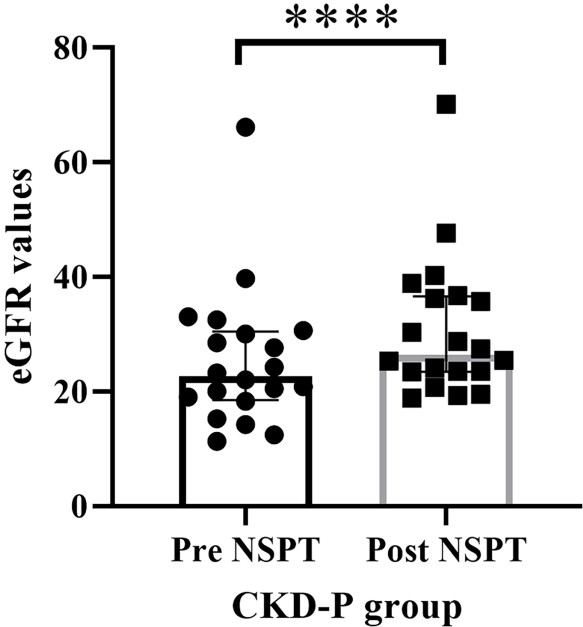

**Figure 4 The improvement in the eGFR values following NSPT.** Asterisks (****) represent $P < 0.0001$. The CKD-P group consists of 20 participants and the Wilcoxon's test was carried out.

(equivalent to 6 weeks). The MLR model including all predictors ($\Delta$PPD, $\Delta$CAL, $\Delta$GBI, $\Delta$PS, $\Delta$IL-6, and $\Delta$TGF-$\beta$1) was not statistically significant ($R^2 = 0.3001$, $P > 0.05$), indicating that these variables did not significantly explain the variation in the rate of eGFR change.

## DISCUSSION

The current study explored the outcomes of NSPT in modulating periodontal parameters, renal function, and concentration of inflammatory biomarkers (IL-6, and TGF-$\beta$1) in CKD patients with periodontitis. The renal function profiles of the three groups were assessed to determine their kidney function. In general, the sodium level in adults should be within 135–145 mmol/L. All three groups showed readings of sodium levels within the reference range. However, the slight difference in sodium levels of both CKD-P and P groups, compared to healthy participants, may be linked to the risk of hypernatremia in CKD patients and the effect of periodontitis on slightly increasing the level of sodium.

*Anuradha et al. (2015)* highlighted that the analysis of saliva samples from CKD patients showed elevated levels of urea, potassium, and phosphate. In agreement with *Anuradha et al. (2015)*, the current study found that the CKD-P group had the highest serum urea and potassium levels compared to the P and HP groups despite using different types of samples. Elevated serum urea level provides a more significant concentration gradient, which promotes urea diffusion from serum to saliva. The prevalence of dental caries and calculus deposition may be due to the elevated amount of urea in CKD patients' saliva (*Anuradha et al., 2015*).

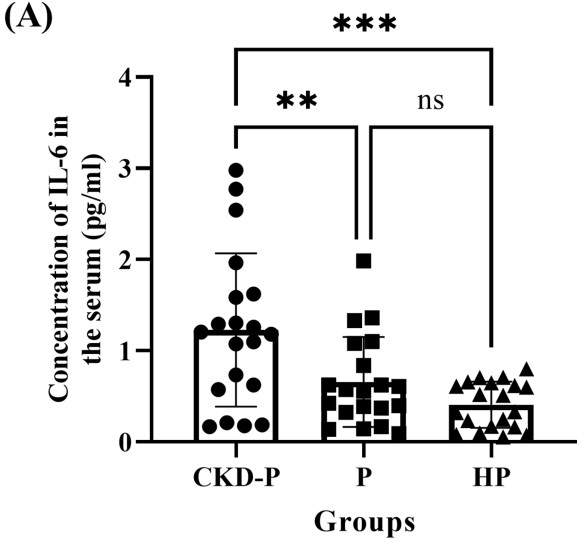

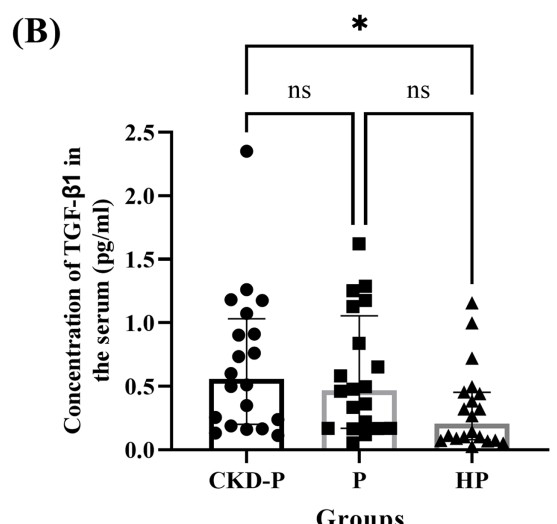

**Figure 5  The comparisons in the concentrations of (A) IL-6 and (B) TGF-β1 in the serum of CKD-P, P, and HP groups.** ns represents non-significant difference, ***$P < 0.001$, **$P < 0.01$ and *$P < 0.05$. Each group consists of 20 participants and the data analysed were measured pre-NSPT. The (A) one-way ANOVA and (B) Kruskal-Wallis analyses were conducted accordingly.

The reference range for serum potassium level is 4.2–4.4 mmol/L. The CKD-P group showed elevated serum potassium levels, while the P and HP groups showed normal readings within the reference range. According to *Yamada & Inaba (2021)*, the risk of developing hyperkalaemia increases proportionally with reduced renal function in CKD patients. Furthermore, the CKD-P group had the lowest calcium level relative to the P and HP groups. According to *Janmaat et al. (2018)*, lower serum calcium is related to a more rapid CKD progression and indicates vitamin D deficiency.

**(A)** 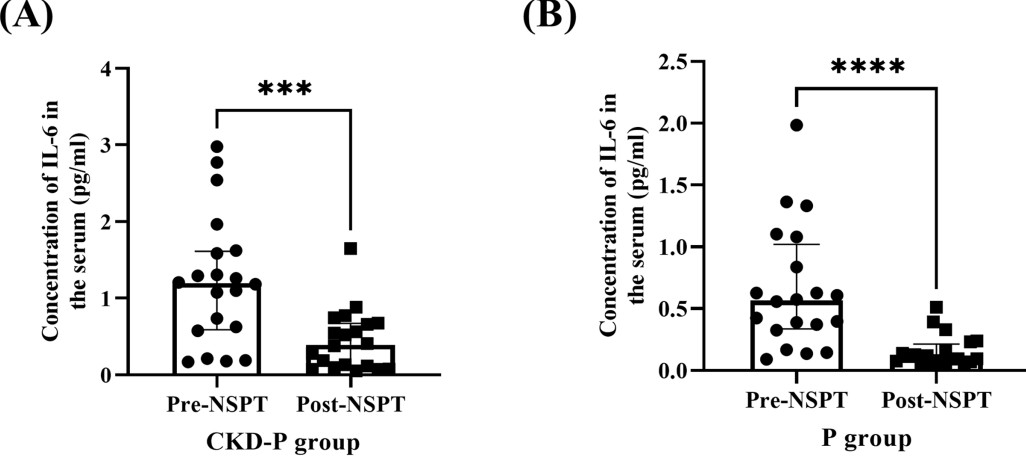 **(B)**

**(C)** 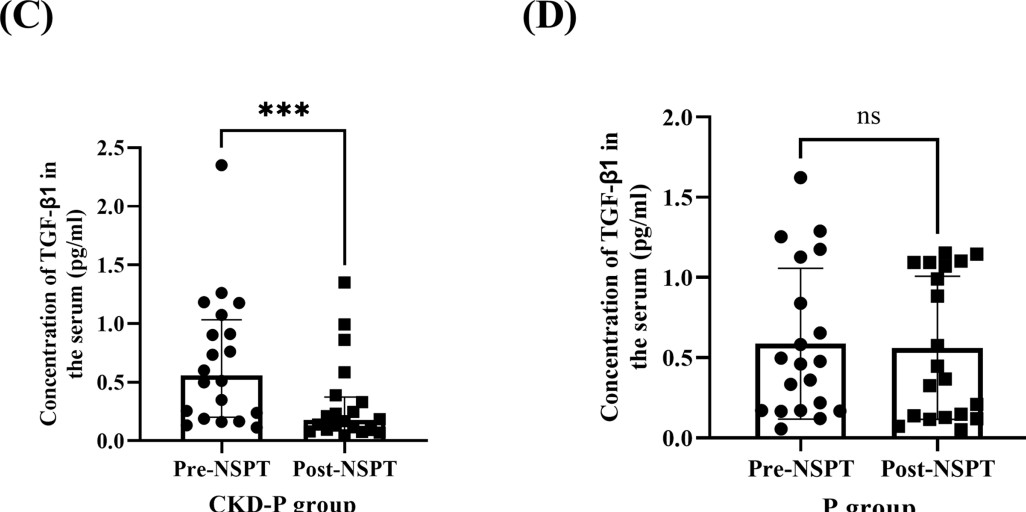 **(D)**

**Figure 6 The serum IL-6 concentration significantly decreased in (A) CKD-P and (B) P groups, followed by a reduction in TGF-β1 in (C) CKD-P, post-NSPT. However, no significant reduction in TGF-β1 was observed in (D) P group, post-NSPT.** ns represents a non-significant difference, ***$P < 0.001$, ****$P < 0.0001$. Both CKD-P and P groups consist of 20 participants. For analyses (A–C), the Wilcoxon's test was carried out whereas for analysis (D), the paired T-test was conducted.

CKD typically worsens over time and is linked to a consistent decrease in kidney function, as reflected by the decline in eGFR and elevated levels of serum creatinine (*Dumnicka et al., 2022*). The eGFR value is determined based on serum creatinine measurement, age, gender, and ethnicity. The normal range of serum creatinine is 62.0–106.0 µmol/L for males and 44.0–80.0 µmol/L for females. The CKD-P group showed the highest serum creatinine level compared to the P and HP groups.

The CKD-P group scored the highest PS compared to the P and HP groups. However, the differences in the PPD, CAL, and GBI were not significant between the groups. Several studies reported that CKD was associated with higher probabilities of attachment loss advancement, increased prevalence of periodontal pockets, and deposition of plaque (*Chaudhry et al., 2022*; *Mahendra et al., 2022*; *Munagala et al., 2022*). According to a recent study by *Ma et al. (2021)*, patients with end-stage renal disease (ESRD) had significantly elevated PPD, GBI, and calculus index than healthy controls. *Chaudhry et al. (2022)* reported that patients with both CKD and periodontitis exhibited more severe periodontal disease compared to patients with only periodontitis.

Antidiuretic drugs consumed by CKD patients may impair the salivary flow, resulting in decreased lubrication and promoting plaque accumulation (*Ibrahim et al., 2020*). Uremic patients have been discovered to have more dental complications in the oral mucosa, teeth, and salivary glands (*Bots et al., 2007*) compared to healthy controls, which appear to develop before dialysis. According to a systematic review and meta-analysis by *Serni et al. (2023)*, the prevalence of periodontitis in CKD patients varied from 34.35% to 93.65%. *Ibrahim et al. (2020)* suggested that CKD could be a risk factor for periodontal disease. Thus, it is crucial to enhance the oral healthcare practice of CKD patients and supervise their oral health condition. The introduction of NSPT improved the clinical outcomes of the periodontal parameters among patients in the CKD-P and P groups. This improvement in the periodontal parameters due to NSPT is parallel with the findings from *Chaudhry et al. (2022)*.

Poor creatinine clearance in the CKD-P group is consistent with their low eGFR reading during pre-NSPT. In line with our earlier study (*Chaudhry et al., 2022*), post-NSPT showed improvement in eGFR value in CKD-P patients. Research has discovered that periodontal treatment significantly improves the eGFR value in CKD patients (*Artese et al., 2010*; *Graziani et al., 2010*). Similarly, *Almeida et al. (2016)* reported significant improvement in eGFR following periodontal treatment (subgingival scaling and root planing), followed by a substantial reduction in asymmetric dimethylarginine (ADMA) level. ADMA is an endogenous inhibitor of NOS, which is elevated in periodontitis.

Cytokines, including IL-1α, IL-1β, IL-6, and TNF-α, play significant functions in the periodontal inflammatory process. In the current study, the CKD-P group exhibited a significantly higher concentration of IL-6 compared to the P group. *Chaudhry et al. (2022)* had previously reported similar findings. In a recent systematic review, studies reported a statistically significant downregulation in IL-6 levels in patients after receiving non-surgical periodontal therapy (*Winiati et al., 2024*). Based on these findings, we can postulate that IL-6 contributed by periodontitis may build up within the body due to renal excretion impairment caused by worsening renal function. However, in the presence of only periodontitis, there was no significant difference in the concentration of IL-6 compared to healthy control. This could imply that the presence of IL-6 is enhanced in patients with both CKD and periodontitis. As a result, compared to the other two groups, CKD-P is the most prominent representative of the pro-inflammatory burden of the systemic condition.

In our study, the CKD-P group exhibited the highest concentration of TGF-β1, as it is produced by kidney cells to assist in the resolution of inflammation and modulate immune cell activity in CKD progression (*Tang et al., 2021*). The significant difference in the TGF-β1 concentration between CKD-P and HP groups (without periodontitis) further validates the more crucial roleplay of TGF-β1 in CKD progression rather than in periodontitis. Potentially, the biological activities of TGF-β1 could have caused inadequate remodelling and perfusion of tooth-supporting tissues, leading to periodontal destruction (*Matarese et al., 2012*).

NSPT has been revealed to minimise the systemic inflammatory burden in patients with CKD, particularly those undergoing haemodialysis therapy. As the periodontal parameters were improved following post-NSPT, there was a significant reduction in the serum concentration of IL-6 in CKD-P patients. This result suggests that the introduction of NSPT may aid in controlling systemic inflammation and reduce the renal function burden for the excretion of IL-6. According to *Sharma et al. (2021)*, a 10% increase in periodontal inflammation leads to a 3% reduction in renal function. As for clinical relevance, reduction in IL-6 level following NSPT may correlate with decreased systemic inflammation, potentially mitigating the progression of CKD. *Dou et al. (2024)* revealed that *P. gingivalis* LPS-stimulated macrophages express hepcidin through the IL-6/STAT3 pathway and undergo polarisation to exhibit M1 phenotype. Hepcidin, a dominant regulatory molecule of iron metabolism, is elevated through the IL-6/STAT3 signalling pathway in inflammatory states (*Agoro & Mura, 2016*). *Dou et al. (2024)* added that the inhibition of the IL6/STAT3 pathway suppressed these changes. *Chen et al. (2019)* postulated that blocking the IL-6 signalling protects against renal fibrosis by suppressing STAT3 activation. Clinically, renal fibrosis is a hallmark of CKD and plays a central role in its progression (*Tang et al., 2021*).

Such a reduction of serum concentration of IL-6 was also observed in the P group. A similar finding was highlighted by *Reis et al. (2014)*, where the total levels of IL-1α, IL-1β, and IL-6 in the gingival cervical fluid (GCF) of disease sites in chronic periodontitis patients reduced in response to non-surgical therapy. The findings of the previous study by *Reis et al. (2014)* are comparable to the present study since there is a proven relationship between serum and GCF (*Lu et al., 2022*). Periodontal health in CKD patients with periodontitis is typically worse in those with periodontitis alone. Reduction in IL-6 post-NSPT suggests a systemic anti-inflammatory effect, which is especially clinically relevant for CKD-P patients as there are potential confounders that could worsen periodontitis amongst CKD patients. The potential confounders may include the older age group associated with CKD and the duration of CKD itself which could further accumulate uremic toxins.

The intervention of NSPT in the CKD-P group lowered the concentration of TGF-β1. According to *Vikram et al. (2015)*, NSPT significantly reduced the concentration of TGF-β1 in the GCF of periodontitis patients. The TGF-β/SMAD signalling pathway stands as a main contributor to renal fibrosis in CKD (*Li et al., 2024*). Persistent TGF-β1 signalling activation can cause chronic inflammation by amplifying the inflammatory responses in CKD (*Tang et al., 2021*). The findings from the current investigation and the study by

*Vikram et al. (2015)* further solidify the role of NSPT in controlling the progression of inflammation in patients with CKD and periodontitis. An *in vivo* study by *Chen et al. (2021)*, reported that periodontitis complicated with obesity contributed to the inflammatory response (increase in cytokines such as IL-6) in mice, which could be alleviated by downregulating the TGF-β/SMAD pathway. However, NSPT did not significantly reduce the level of TGF-β1 in the P group, which may further emphasise the relevant role of TGF-β1 in CKD (*Tang et al., 2021*) rather than in periodontitis. In a study by *Aukkarasongsup et al. (2013)*, treatment with periostin inhibited TGF-β1/SMAD signalling by blocking the TGF-β1 receptor under hypoxic conditions. Periostin is an extracellular matrix protein expressed predominantly in periodontal ligament cells. However, periostin's regulatory effects on TGF-β1 signalling are specific to the hypoxic conditions and may not be as prominent under normal oxygen levels.

Overall, we postulate that NSPT may not only treat periodontal parameters but also improve systemic inflammation *via* the reductions in IL-6 and TGF-β1 and subsequently enhance renal function. Such occurrences are believed to be achieved *via* the interaction between various molecular signalling pathways such as IL6/STAT3 and TGF-β1/SMAD. Improvement in systemic inflammation may enhance renal function as these pathways are also involved in the CKD progression.

Aside from that, we have also attempted an MLR analysis to distinguish whether modifications in the dental parameters and inflammatory biomarkers may predict the improvements in eGFR. However, the outcomes were insignificant. The analysis could be repeated in the future using a larger sample size.

This study provides valuable insights into the relationship between periodontitis and CKD by evaluating IL-6 and TGF-β, two key cytokines involved in inflammatory and fibrotic pathways. While previous research has largely focused on systemic inflammation markers such as CRP, our study highlights the dual role of inflammation and fibrosis in CKD pathophysiology among periodontitis patients. This approach enhances the understanding of disease progression and reinforces the importance of oral health in systemic disease management. The identification of IL-6 and TGF-β1 as potential biomarkers may contribute to early detection, risk stratification, and the development of personalised treatment strategies in both periodontal and nephrology care. Furthermore, the study employs a rigorous methodology, including a well-defined patient selection process and standardised cytokine measurement techniques, ensuring the reliability and validity of findings.

However, the study has several limitations, including a small sample size and short follow-up duration, which may limit the generalisability and long-term implications of the results. A larger cohort would assist in confirming whether the observed findings in the present study, especially in periodontal parameters and inflammatory markers are consistently applicable across diverse populations. A shorter follow-up period may not fully capture the long-term effects of NSPT on renal function and inflammatory markers. An extended follow-up period (3–6 months) (*Artese et al., 2010*; *Graziani et al., 2010*; *Fang et al., 2015*) is required to evaluate the lasting impact of NSPT and to confirm whether initial improvements in periodontal parameters, levels of inflammatory markers and renal

function are maintained over time. Additionally, the single-centre design and lack of control for lifestyle factors, such as diet and physical activity, may introduce biases and restrict external validity. While the study provides significant associations, the absence of mechanistic exploration leaves room for further molecular investigations.

## FUTURE DIRECTION

To deepen the understanding of the CKD-periodontitis-inflammation relationship, it is essential to conduct larger, multicentre cohorts with extended follow-up periods. This investigation should incorporate molecular analyses to elucidate the pathways underlying this complex interaction. Additionally, more comprehensive adjustments for potential confounding factors such as detailed dietary assessments, medication logs, and lifestyle questionnaires are crucial to minimise bias and control for variables that may impact the results. Future studies should also consider including dialysis patients as dialysis may interfere with the systemic inflammation profile and influence periodontal outcomes. To further validate the systemic health improvements following NSPT, additional inflammatory biomarkers such as IL-8 and IL-1β (*Teles et al., 2024*), oxidative stress markers, kidney function indicator such as cystatin c, and imaging techniques could be employed. These approaches will provide a more robust understanding of the effects of NSPT on both periodontal and renal health.

## CONCLUSIONS

In conclusion, this study demonstrates that NSPT significantly improves periodontal parameters, enhances renal function, and reduces systemic inflammation in CKD patients with periodontitis, thus the hypothesis is accepted. The findings underscore the bidirectional relationship between periodontal health and systemic conditions, particularly in CKD, where NSPT effectively reduces inflammatory biomarkers such as IL-6 and TGF-β1 while enhancing eGFR. These results highlight the potential of NSPT as an adjunctive therapeutic strategy in the multidisciplinary management of CKD, emphasising the importance of integrating dental care into systemic disease management. However, the findings also call for further research involving larger, multicentre cohorts and extended follow-up periods to validate these outcomes and explore the molecular mechanisms underpinning the observed improvements. Integrating NSPT into routine CKD management protocols could have significant implications for improving patient outcomes, reducing the systemic inflammatory burden and delaying disease progression. Despite the potential benefits, several barriers may hinder the widespread adoption of periodontal therapy in CKD management, such as lack of awareness and multidisciplinary collaboration.

## LIST OF ABBREVIATIONS

**CAL**  Clinical attachment loss
**CEJ**  Cement-enamel junction
**CKD**  Chronic kidney disease
**CRP**  C-reactive protein

| eGFR | Estimated glomerular filtration rate |
|---|---|
| GBI | Gingival bleeding index |
| ICC | Intraclass correlation coefficient |
| IL-6 | Interleukin-6 |
| IQR | Interquartile range |
| MLR | Multiple linear regression |
| NOS | Nitric oxide synthase |
| NSPT | Non-surgical periodontal therapy |
| PPD | Periodontal pocket depth |
| PS | Plaque score |
| ROS | Reactive oxygen species |
| SD | Standard deviation |
| SRP | Scaling and root planing |
| TGF-$\beta$1 | Transforming growth factor-beta 1 |
| TLRs | Toll-like receptors |
| TNF-$\alpha$ | Tumour necrosis factor alpha |

### Funding
This work was supported by the Fundamental Research Grant Scheme (FRGS/1/2022/STG02/USM/03/1), Ministry of Education, Malaysia. The funders had no role in study design, data collection and analysis, decision to publish, or preparation of the manuscript.

### Grant Disclosures
The following grant information was disclosed by the authors:
Fundamental Research Grant Scheme, Ministry of Education, Malaysia: FRGS/1/2022/STG02/USM/03/1.

### Competing Interests
The authors declare there are no competing interests.

### Author Contributions
- Harishini Rajaratinam performed the experiments, analyzed the data, prepared figures and/or tables, authored or reviewed drafts of the article, and approved the final draft.
- Nurul Aliya Abdul Rahman performed the experiments, analyzed the data, prepared figures and/or tables, authored or reviewed drafts of the article, and approved the final draft.
- Muhammad Hafiz Hanafi conceived and designed the experiments, authored or reviewed drafts of the article, and approved the final draft.
- Siti Lailatul Akmar Zainuddin performed the experiments, analyzed the data, authored or reviewed drafts of the article, and approved the final draft.

- Hanim Afzan Ibrahim performed the experiments, analyzed the data, authored or reviewed drafts of the article, and approved the final draft.
- Muhammad Imran Kamarudin performed the experiments, analyzed the data, authored or reviewed drafts of the article, and approved the final draft.
- Wan Mohd Saifuhisam Wan Zain performed the experiments, analyzed the data, authored or reviewed drafts of the article, and approved the final draft.
- Sirajudeen Kuttulebbai Nainamohamed Salam conceived and designed the experiments, authored or reviewed drafts of the article, and approved the final draft.
- Salbiah Isa performed the experiments, authored or reviewed drafts of the article, funding acquisition, and approved the final draft.
- Nur Karyatee Kassim conceived and designed the experiments, authored or reviewed drafts of the article, supervision, Resources, Project administration, and approved the final draft.

### Human Ethics

The following information was supplied relating to ethical approvals (*i.e.*, approving body and any reference numbers):

Human Research Ethics Committee of Universiti Sains Malaysia

### Data Availability

The raw data are available in the Supplemental File.

### Supplemental Information

Supplemental information for this article can be found online at http://dx.doi.org/10.7717/peerj.19492#supplemental-information.

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
