# Peer review of "Exploring the outcomes of non-surgical periodontal therapy in modulating periodontal parameters, renal function, and inflammatory biomarkers in chronic kidney disease patients with periodontitis"

_PeerJ, doi:10.7717/peerj.19492_

## Round 0.1 · original submission · Major Revisions

Both reviewers found that the study has merit, but various revisions can significantly enhance the level of information presented in the manuscript. Please address all reviewers' concerns carefully by thoroughly revising your manuscript. Additionally, Figures 1–5 should present the data in a more informative manner than with error bars. Box plots could be a valid alternative.

Reviewer 1 ·

Basic reporting

these areas are acceptable

Experimental design

see comments

Validity of the findings

see comments

Additional comments

Dear Authors ,
Congratulations on your well conducted study. I have few queries and recommendations to improve the presentation-
In introduction- Please add more relevant and recent literature to support the rationale for investigating non-surgical periodontal therapy (NSPT) in CKD patients.; Elaborating on specific pathways involving IL-6 and TGF-β.; Describe why previous studies have failed to address the combined focus on periodontal health and renal function improvements through NSPT.; Provide a detailed rationale for selecting IL-6 and TGF-β1 as inflammatory markers while excluding others like CRP or TNF-α.

In methods- Clarify the procedure for measuring periodontal parameters (e.g., describe the calibration process for inter-examiner reliability).; Detail the rationale for the six-week follow-up period—why this duration was deemed sufficient to observe meaningful changes.; Include a justification for the sample size. A power analysis would strengthen the methodological rigor.; What are the views on other potential confounding variables (e.g., diet, medication compliance) and how these were controlled.; Improve figure readability by increasing font size and simplifying legends.

In discussion- Discuss the clinical relevance of observed changes in IL-6 and TGF-β1 levels. How do these reductions correlate with systemic health improvements.; Provide more context on the differences in periodontal health between CKD-P and P groups (e.g., potential confounders such as age or duration of CKD).
Expand on how the findings align or diverge from similar studies. For example, compare the IL-6 reductions observed in this study with those reported in prior systematic reviews.; Discuss potential molecular pathways through which NSPT affects renal function and inflammatory markers.; Highlight how improved periodontal parameters might alleviate systemic inflammation, potentially slowing CKD progression.; Propose more comprehensive investigations (e.g., multicenter studies, inclusion of dialysis patients).

Write in detail the limitations explaining the relevance of each point and how it affected your study. ; Suggest additional biomarkers or imaging studies to validate systemic health improvements.; Emphasize the need for integrating periodontal therapy into CKD management protocols and outline potential barriers to implementation.

Add a table for abbreviations. rewrite the conclusion specifically answering the aim of the study.
write your hypothesis and in discussion justify its acceptance or rejection.

Reviewer 2 ·

Basic reporting

This study is a cohort study to investigate the effect of non-surgical periodontal therapy on periodontal outcomes (periodontal pocket depth and clinical attachment loss) and kidney function (glomerular filtration rate) in patients with combined chronic kidney disease and periodontitis or periodontitis only, using healthy volunteers as controls.

Strengths of the study include experimental design (recruitment of appropriate participants, measurement of relevant biological parameters, description of methods). Weaknesses of the study include presentation of data, analysis of data, reporting of the data, and discussion of the reported data, particularly regarding the underlying association between periodontitis and chronic kidney disease. However, I think these weaknesses can be addressed in a revision of the manuscript. Below I have provided recommendations to help the authors improve their manuscript.

Basic reporting

Introduction: Overall, the introduction provides insufficient background to the problem. Key supporting references are quite dated (e.g.Vilela et al., 2011), or missing (e.g lines 82-84).

In lines 85-95, a description of how a periodontal pathogen, Porphyromonas gingivalis, impacts cardiovascular health is provided. While bacteria or their products can influence kidney health via changes to the cardiovascular system, the relevance to the current study is unclear. No cardiovascular outcomes were measured in the present study. Therefore, any reference to endothelial dysfunction as a link between periodontitis and CKD should be minimum in the introduction of this study (or could be moved to the discussion as an area for future research).

Instead, the context and background to the problem could be improved by highlighting research that supports a bi-directional relationship between periodontitis and CKD. What evidence is already known? What are the current limitations? How does the present study aim to reproduce/extend/advance current knowledge?

The rationale for assessing IL-6 and TGF-beta is not clear. Two references are provided (Zhao et al., 2019; Oza et al., 2024). However, neither reference provides evidence or recommends measuring IL-6 or TGF-beta as mediators between the two conditions. However, TGF-B beta is implicated in CKD, and IL-6 is implicated in periodontitis and CKD. Therefore, the rationale for selecting these biological parameters in the present study can be described – and cited appropriately. Further, it will be important to indicate why these cytokines were selected over the more traditional CRP.

Experimental design

Overall, the experimental design is appropriate and well-described.

Additional details regarding ethical approval are required in the methods section, including the name of the granting organisation and approval reference numbers.

Suggested improvements to the analysis and presentation of data is described in the section ‘Validity of findings’.

Validity of the findings

Table 1: A statistical difference in age-distribution is highlighted. Please use the figure legend to describe the post-hoc test used to identify between which groups the statistical difference lies, and indicate this on the table.

Table 2: A table is provided, whereby P-values derived from one-way ANOVA tests are provided. It is currently unclear whether this P-value is the overall P value for the test, or whether the result of Post-hoc tests to determine where the statistical difference exists. Please indicate in the table which comparisons this P-value refers. For example is this the difference between CKF-P patients and healthy controls, after post-hoc tests?

Further, as these measurements relate to baseline measurements in CKD-P patients, it would be preferable to present the table and discuss these findings before the pre vs post-NSPT are presented/discussed.

Clinical periodontal parameters: In several instances small, non-significant statistical differences are described as a ‘higher mean’ in the abstract, lines 241, 242, and in the discussion. In each instance this should be changed to say ‘there was no evidence of a difference in parameter [X] between CKD-P and P patients’ (or similiar).

In the interest of transparency – and to enable readers to see the actual spread of the data it would be preferable to change the data from solid bars with SD to show the actual data points with the mean and SD. Such a simple change to the data presentation is easily achieved with GraphPAd prism. Further, in each figure the legend should contain information relating to the number of participants in each group, the statistical tests used for each analyses, and whether the data are pre-NSPT measurements (e.g. figure 1), or pre vs post-NSPT measurements (e.g Figure 5).

Changes in the glomerular filtration rate pre- and post-NSPT are not presented (and are instead simply described in the text). This is surprising, as these data provide evidence that NSPT has a biological effect on kidney function. I would have expected these data to be the primary outcome measure for the study.

In Figure 6 a correlation analyses is performed to associate reductions in GFR pre-and post-NSPT with reductions in IL_6 and TGF-beta. However, the authors should consider instead conducting a multiple-regression analyses to measure how NSPT (using changes in periodontal outcomes) contributes to improvements in GFR. Such an analysis could reveal whether NSPT is effective at improving kidney function in patients with CKD. If necessary the authors should consider consulting a statistician to perform this analyses, which would significantly extend the meaningful value of the study.

Discussion: The discussion should be reworked in light of the suggested changes. Please make sure to place the findings of this research in context of the link between CKD and periodontitis. Highlighting how this study extends/compliments/contradicts existing evidence, and highlighting the weaknesses/strengths of the revised manuscript.

---

## Round 0.2 · Minor Revisions

Dear authors,

Please review your manuscript according to the reviewer's comments.

Reviewer 1 ·

Basic reporting

acceptable

Experimental design

acceptable

Validity of the findings

acceptable

Additional comments

not required

Reviewer 2 ·

Basic reporting

Several revisions have been made to the background information provided in the introduction, in line with previous feedback of the initial submission. These changes include a clearer rationale for the assessment and relevance of IL-6 and TGF-beta as outcome measures in this study. Further, a substantial proportion of the introduction highlighting endothelial dysfunction as a potential mechanistic link was removed as this was not deemed relevant to the current study design and outcome measures.

Unfortunately, evidence supporting a bidirectional relationship still has not been adequately established in the manuscript. Lines 83-114 discuss proposed mechanistic links between periodontitis and CKD. What is remains missing is the presentation of epidemiological evidence supporting a clinical bi-directional relationship between periodontitis and CKD. This is important to establish since the hypothesis presented in the present study advocates for NSPT as a potential clinical intervention to enhance renal function and reduce inflammation in patients with CKD.

Lines 85-88 have been copied verbatim, including an in-text citation, from another publication (Baciu et al, 2023). Further, this review article is cited as evidence that P. gingivalis LPS activates TLR2 and TLR4 in periodontal tissues, and in the kidneys (lines 90-93). This is misleading and incorrect. There is evidence that P. gingivalis LPS can trigger TLR signalling in human embryonic kidney cells in vitro. Further, it is widely accepted that P. gingivalis expresses an atypical LPS that antagonises TLR4 activation doi: 10.1111/j.1462-5822.2009.01349.x .

Line 102 ‘Excessive inflammation, stimulated by the escape mechanisms leads to oxidative stress.’ What is meant by escape mechanisms – and how is this relevant to the aims of the current study? Further, it is unclear how the additional background information around oxidative stress (lines 102-110) is relevant to the aims of the current study.

Periodontitis and CKD are separate conditions, and are necessarily treated as such. Please revise lines 128-129.

I urge the authors to revise the introduction to report only the basic information required to understand the need, potential benefit and purpose of the current study.

Requested changes to the presentation of figures and tables have been implemented. These changes will enhance the readers assessment of the presented data.

A multiple linear regression was performed. There was potential here for the authors to use this analysis (or other relevant analysis) to use their existing dataset to measure the treatment effect of NSPT intervention on improvements in the eGFR. This could have been performed in a variety of ways, either using the coefficient of eGFR from baseline to follow-up as the dependent variable for MLR or using ANCOVA to assess changes in eGFR from baseline to follow-up. However, the analysis presented instead measures the relationship between post-NSPT parameters with post-NSPT eGFR. Consequently, this analysis does not measure the effect of NSPT intervention on improvements in eGFR. While it is clear from line 364 what data were used in the current analysis – the limitation or using only post-NSPT data should be highlighted, or the analysis revised as suggested.

Experimental design

no comment

Validity of the findings

no comment

---

## Round 0.3 · accepted · Accept

Thank you for the revisions to the manuscript.